# Sparse Hyperspectral Band Selection Based on Expectation Maximization

## Abstract

Band selection is crucial in spectral imaging, as it involves choosing the most relevant bands from large hyperspectral datasets to retain essential information while reducing the burden of data transmission and analysis. Addressing this need, we introduce a novel method for band selection that utilizes an Expectation Maximization algorithm to facilitate selection through the sparsification of spectral band importance. Our method enhances sparsity effects and effectively delineates the relationships between spectral bands during the sparsification process. Supported by thorough theoretical analysis and experimental validation on public datasets, our approach has proven to be both robust and practical. Compared to other sparsification methods, it not only excels in achieving significant sparsity effects but also demonstrates marked advantages in illustrating inter-band relationships. Our method delivers outstanding performance in band selection tasks and holds potential for broader applications in other sparsity-oriented contexts in the future.

## 1 Introduction

Hyperspectral imaging technologies, extensively employed in areas like remote sensing and agriculture, provide detailed spectral information but encounter challenges such as high operational costs and complex data processing Mahdianpari et al. (2018); Toker et al. (2022); Zhang et al. (2022); Xiao & Wei (2023). A fundamental aspect in this domain is the efficient selection of the most informative spectral bands from the extensive data available. This selection is not only crucial for enhancing classification performance, as demonstrated by the Hughes phenomenon, but also for reducing data transmission stress, thereby making the technology more adaptable for various applications.

Recent advancements in band selection, particularly those utilizing deep learning techniques Zhou et al. (2023); Sun et al. (2021), mark a significant shift away from traditional methods Tang et al. (2021); Huang & He (2005); Zhang et al. (2007); Fauvel et al. (2015). These modern approaches, which include reconstruction-based Cai et al. (2019) and classification-based methods Feng et al. (2020); Jia et al. (2023), are primarily focused on learning the importance of spectral bands. The central challenge lies in designing mechanisms that effectively represent band importance in alignment with these specific tasks. The importance extends beyond the contribution of individual bands, encompassing the significance of various band combinations. For instance, NDVI and NDWI are commonly used empirical formulas in spectroscopy that assess material properties based on the interactions among different spectral bands.

However, it has been noted that many existing methods fall short in effectively capturing the importance of relationships between spectral bands. The two most prevalent techniques, clustering and ranking, both necessitate post-processing, namely the selection of bands based on their relative importance. These approaches are problematic because the assigned importance is not always precise and overlooks the interplay between bands, potentially excluding crucial band combinations. To overcome these limitations, we have adopted a method based on the sparsity of band importance, leveraging the training process of the network to inherently select bands. This method aims to enable the network to independently discover the most representative bands, thereby circumventing the conventional need for post-processing. However, this approach introduces new challenges: 1) Existing methods of imposing sparsity, such as L1 and L2 losses, do not consistently yield stable sparsity effects; 2) Current sparsity techniques also fail to depict the inter-band relationships during the sparsification process.

To address these challenges, we have developed innovative solutions within our framework. We introduce a Sparsity Loss based on the Expectation Maximization (EM) algorithm Dempster et al. (1977), meticulously designed to enhance sparsity in the levels of importance. This method not only improves sparsity effects but also theoretically facilitates the exploration of relationships between spectral bands, thereby enabling more stable and reliable band selection. To our knowledge, our approach is the first to implement a sparsity representation method based on the EM algorithm. We have conducted comprehensive theoretical analysis and experimental validation to confirm the sparsity effects and the capability of our proposed method to accurately depict inter-band relationships. Our approach has undergone rigorous analysis and has been thoroughly validated.

In summary, our main contributions are as follows: 1) We have developed a band selection method for deep learning that utilizes sparse importance representation, suitable for both supervised and unsupervised tasks. The integration of Sparsity Loss effectively alleviates the challenge of depicting the importance among spectral bands. 2) Our methodology introduces an innovative and unique sparsification technique based on the Expectation-Maximization algorithm, marking a significant advancement in the field of sparse representation. 3) This paper supports our proposed method with comprehensive theoretical proofs and analyses, alongside extensive experimental evidence to validate the efficacy and benefits of our approach. Our method achieves state-of-the-art performance in band selection methods and, in significance tests, significantly outperforms other sparsity methods.

## 2 RELATED WORK

The classification of traditional band selection methods for hyperspectral images, as outlined by Sun et al. Sun & Du (2019), is organized into six categories: Ranking methods (Chang et al. (1999); Huang & He (2005)) assess band importance based on predefined criteria but ignore correlations among bands. In contrast, search-based methods (Zhang et al. (2007); Fauvel et al. (2015)) optimize subsets while considering these correlations, albeit at a higher computational cost. Clustering methods (Qian et al. (2009); Imbiriba et al. (2015)) reduce redundancy by selecting representative bands from clusters, though their effectiveness can depend on the chosen algorithm. Sparsity methods (Sun et al. (2015; 2017)) utilize sparse representation for band selection, facing challenges in parameter optimization. Embedding learning (Zhang & Ma (2009); Zhan et al. (2017)) combines selection with classifiers for end-to-end optimization but may lack interpretability. Lastly, hybrid schemes (Datta et al. (2015); Jiang & Li (2015)) blend various approaches to capitalize on their strengths, resulting in enhanced effectiveness but increased complexity.

With the rapid development of deep learning Shone et al. (2018); Zhang et al. (2021); Pramanik et al. (2021), band selection methods based on deep learning Ribalta Lorenzo et al. (2020); Feng et al. (2020); Pestel-Schiller et al. (2021); Yu et al. (2022) have garnered significant attention in the academic community. These methods are recognized for their ability to automatically learn complex features from hyperspectral data and effectively handle high-dimensional, highly correlated, and noisy data, adapting to various application scenarios. For instance, Cai et al. (2019) introduced BS-Nets, which use attention mechanisms to assess band importance, while Zhao et al. (2020) enhanced classification performance through CNN interpretability using 1D GradCAM to visualize band contributions. Wu & Yan (2021) developed HyperDesc for joint optimization of band selection and feature extraction via a non-local spectral-spatial attention network. Most recently, Zhou et al. (2023) proposed IGAEBS, an unsupervised hyperspectral band selection method that uses GCNs to extract structural features and iteratively refines a band relation graph. While Yao et al. (2024) designed a novel BS module and cascaded band-specific spatial attention blocks for supervised band selection. However, these methods do not fundamentally address the issue of spectral band confidence representation. To tackle these challenges, we introduce a novel sparse learning strategy that achieves band selection in one step while implicitly representing the relationships between spectral bands.

## 3 METHOD

### 3.1 OVERVIEW

In response to the challenge of band selection, we propose a robust and efficient sparse framework. Our model adopts a task-driven approach to training band selection. The selection process utilizes

importance levels to identify an optimal subset of bands, with selected bands assigned an importance level of 1, and unselected bands marked as 0. The training is conducted in an end-to-end manner, incorporating both task loss and sparsity loss into a deep network that has been randomly initialized for the specific task. After a set number of training iterations, this method effectively achieves a sparse distribution of band importance. The loss function is presented as follows:

$$L = L_{task} + \alpha L_{sp}, \tag{1}$$

where $L_{task}$ represents the loss for a subsequent task, which is the classification loss for supervised image classification tasks, and the reconstruction loss for unsupervised tasks. $L_{sp}$ denotes the sparsity loss, and $\alpha$ is a hyperparameter for adjusting the impact of the sparsity loss. In the following sections, we will elaborate on the principle and the calculation method of $L_{sp}$.

## 3.2 Parameter Definition

Let $c = \{c_i\} = \{c_1, c_2, \ldots, c_B\}$ represent the importance weights of each spectral band, where $c_i \in [0, 1]$; $B$ denotes the number of spectral bands, and $k$ indicates the number of bands to be selected ($k \leq B$). The selection status of the $i$-th band is denoted by $b_i$, where $b_i = 1$ indicates that the $i$-th band is selected, and $b_i = 0$ means it is not selected. Each possible selection of spectral bands is represented by $\pi$ (e.g., $\pi = [1, 0, 1, 0, 0, 1, 0, ...]$), which is a vector of $1/0$ values of length $B$, indicating whether each band $i$ is selected ($b_i = 1$ or $b_i = 0$) under a particular selection $\pi$. $b_i = Sign(\pi, i)$ ($Sign(\pi, i) \in \{0, 1\}$) indicates the selection status $b_i$ of band $i$ in a given selection $\pi$. $S_{(k,B)}$ represents the event of sparse selection of $k$ bands from $B$ bands. $C_{(k,B)}$ denotes the set of all possible selections $\pi$ for selecting $k$ bands from $B$ bands.

For the raw spectral data input $\mathbf{X}$, we first multiply each spectral band by its corresponding importance weight $c_i$ to obtain $\mathbf{X'}$,

$$x'_{i,h,w} = x_{i,h,w} \cdot c_i, \quad i = 1, 2, 3, \ldots, B, \tag{2}$$

where $x'_{i,h,w}$ represents a spectral value in $\mathbf{X'}$ in the $i$-th channel at the $(h, w)$ location. $\mathbf{X'}$ is used as the input for the network corresponding to subsequent tasks to compute $L_{task}$. The importance weights $c_i$ are used to calculate $L_{sp}$, to achieve sparsification of $c_i$.

## 3.3 Sparse Loss Based on the EM Algorithm

We have designed a band importance sparsity method based on the EM algorithm. In the E step, the probability sum of all selections of $k$ spectral bands from $B$ bands ($\pi \in C_{(k,B)}$) is calculated to obtain the expected value $E_{(k,B)}$, which is used to determine $L_{sp}$. The M step involves minimizing $L_{task} + \alpha L_{sp}$ through each iteration of gradient backpropagation. This step enables the training of the subsequent task model while achieving a sparse selection of the spectral bands.

**E Step:** Calculate the probability sum of all possible selections selecting $k$ spectral bands, which is $E_{(k,B)}$. The derivation process for $E_{(k,B)}$ is as equation 3,

$$
\begin{aligned}
P(S_{(k,B)}|c) &= \sum_{\pi \in C_{(k,B)}} P(S_{(k,B)}, \pi|c) \\
&= \sum_{\pi \in C_{(k,B)}} P(S_{(k,B)}|\pi, c)P(\pi|c) \\
&= E_{\pi \sim P(\pi|c)}[P(S_{(k,B)}|\pi, c)] \\
&= \frac{1}{2^B} E_{(k,B)}
\end{aligned}
\tag{3}
$$

where $P(S_{(k,B)}|c)$ is the probability of sparsely selecting $k$ spectral bands from $B$ bands, and $P(\pi|c)$ represents the prior distribution of the selection $\pi$. We assume that all selections $\pi$ are equivalent, that is, $P(\pi|c) = P(\pi) = \frac{1}{2^B}$, following a uniform distribution; $P(S_{(k,B)}|\pi, c)$ represents the sparse probability given a selection $\pi$,

$$P(S_{(k,B)}|\pi, c) = \prod_{i=1}^{B} p(b_i = Sign(\pi, i)|c), \tag{4}$$

where,

$$p(b_i = 1|c) = c_i$$
$$p(b_i = 0|c) = 1 - c_i. \tag{5}$$

Thus, the calculation of $E_{(k,B)}$ merely involves summing the probabilities of all selections of selecting $k$ bands. For $E_{(k,B)}$, the following relationship holds:

$$\sum_{k=0}^{B} E_{(k,B)} = 1 \tag{6}$$

The specific calculation process is detailed in the following subsection.

**M Step:** Combining the loss from subsequent tasks with the expected value obtained, the model's loss function is formulated to simultaneously achieve the subsequent tasks and parameter sparsity,

$$L_{sp} = -\log E_{(k,B)}, \tag{7}$$

$$L = L_{task} + \alpha L_{sp}. \tag{8}$$

$$c_i^{(t+1)} = c_i^{(t)} - \nabla(L_{task} + \alpha L_{sp}), \tag{9}$$

where $L_{task}$ is the loss associated with the subsequent task, which may either be cross-entropy for classification tasks or mean squared error for reconstruction tasks. Here, $c_i^{(t)}$ denotes the value of $c_i$ following the $t$-th update iteration in the training process.

### 3.4 REASONS FOR ACHIEVING SPARSITY

The ability of our method to achieve sparsity can be elucidated through two theorems:

**Theorem 1:** Within the range of $c_i \in [0, 1]$, $E_{(k,B)}$ assumes values in $[0, 1]$. It achieves a value of 1 if and only if, within the set $\{c_i\}$, $k$ elements are equal to 1 and $(B - k)$ elements are equal to 0.

**Theorem 2:** Within $c_i \in (0, 1)$, $E_{(k,B)}$ has no local maxima but only a saddle point at $c_1 = c_2 = \ldots = c_B = \frac{k}{B}$.

Theorem 1 establishes the upper limit of $E_{(k,B)}$ and demonstrates that it attains its maximum value exclusively under sparsity conditions. Theorem 2 indicates that during the optimization process, one does not encounter any local maxima, implying that the initial values of $c_i$ will most likely eventually converge to the maximum value of $E_{(k,B)}$, which corresponds to a sparse configuration. This theoretically demonstrates the robustness and stability of our method, avoiding convergence to non-sparse local maxima. For the detailed proof, please refer to the appendix.

### 3.5 SOLVING $E_{(k,B)}$ USING DYNAMIC PROGRAMMING

We have constructed a computation graph where solid dots represent a spectral band being selected with a probability $p(b_i = 1|c)$, and hollow dots represent a spectral band not being selected, with a probability $p(b_i = 0|c)$. This visualization is depicted in Figure 1(a). The graph is organized into 2k+1 rows and B columns. The sparsification loss $L_{sp}$ is calculated as the sum of probabilities along paths (selections) that traverse two specific points, marked in a dashed box at the bottom right corner of the graph.

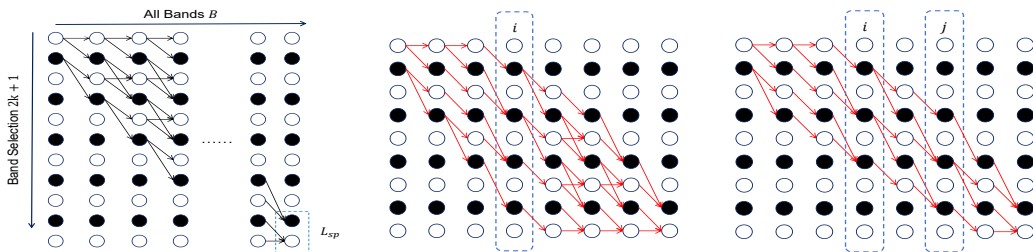

(a) Computation graph for $E_{(k,B)}$ calculation.
(b) Representation of $p(b_i = 1, S|c)$ in the computation graph.
(c) Representation of $p(b_j = 1, b_i = 1, S|c)$ in the computation graph.

Figure 1: Visualizations related to the computation of sparsification loss $L_{sp}$ and probability representations in the computation graph.

The recursive formula is defined as follows:

$$
a_j^i = \begin{cases}
p(b_1 = 0|c) & \text{if } j = 1 \text{ and } i = 1, \\
a_j^{i-1} p(b_i = 0|c) & \text{if } j = 1 \text{ and } i \neq 1, \\
p(b_1 = 1|c) & \text{if } j = 2 \text{ and } i = 1, \\
a_{j-1}^{i-1} p(b_i = 1|c) & \text{if } j = 2 \text{ and } i \neq 1, \\
(a_j^{i-1} + a_{j-1}^{i-1}) p(b_i = 0|c) & \text{if } j = 3, 5, \ldots, 2k+1 \text{ and } i \neq 1, \\
(a_{j-1}^{i-1} + a_{j-2}^{i-1}) p(b_i = 1|c) & \text{if } j = 4, 6, \ldots, 2k \text{ and } i \neq 1, \\
0 & \text{else .}
\end{cases}
\tag{10}
$$

Here, $a_j^i$ represents the cumulative probability of reaching the point (i, j) from the beginning. Upon completing the iterations of the forward algorithm:

$$
E_{(k,B)} = a_{2k}^B + a_{2k+1}^B,
\tag{11}
$$

$$
L_{sp} = -\log(a_{2k}^B + a_{2k+1}^B).
\tag{12}
$$

### 3.6 Advantages in Depicting Relationships Between Spectral Bands

Within our framework, it is possible to describe the relationships between spectral bands. Firstly, unlike other sparsification methods, our approach involves competition among selections $\pi$, which inherently represents the multivariate relationships among spectral bands. Secondly, from the perspective of individual spectral bands, our method can theoretically describe the relationships between bands, specifically, $P(b_j = 1|b_i = 1, S_{(k,B)}, c)$. This probability indicates the likelihood that band $j$ is selected given that the sparse event $S_{(k,B)}$ occurs and band $i$ is selected:

$$
P(b_j = 1|b_i = 1, S_{(k,B)}, c) = \frac{P(b_j = 1, b_i = 1, S_{(k,B)}|c)}{P(b_i = 1, S_{(k,B)}|c)}.
\tag{13}
$$

Here, $P(b_j = 1, b_i = 1, S_{(k,B)}|c)$ denotes the sum of probabilities for all selections that select $k$ spectral bands passing through nodes $b_j = 1$ and $b_i = 1$; $P(b_i = 1, S|c)$ represents the sum of probabilities for all selections that select $k$ spectral bands passing through node $b_i = 1$ (as shown in Figure 1(b) and 1(c)). Similarly, the expression for $P(b_j = 1 \mid b_i = 1, b_k = 1, b_m = 0, \ldots, S_{(k,B)}, c)$ is the same.

Therefore, during the sparsification process, as the importance of a particular band increases, it influences the probability of selection for all spectral bands globally. This means that bands with significant local importance may no longer be decisive and could be influenced by the global distribution of bands, tending towards a relatively optimal global outcome. Our experiments also demonstrated a sequentially sparsified series of attributes, leading us to believe that our method can describe the multivariate relationships between spectral bands. Our future work will continue to focus on addressing this issue, as we believe our theoretical framework provides a viable solution.

## 3.7 ANALYSIS OF $p(\pi|c)$

The physical meaning of $p(\pi|c)$ is that, prior to selecting spectral bands, we assume that all bands or band selections are equivalent. This assumption is commonly made in the design and comparison of spectral band selection algorithms. Importantly, our method is also applicable to specific scenarios or applications with additional requirements for the selected bands. It does not necessitate a globally uniform distribution, rather, a piecewise locally uniform distribution can also be effectively utilized. Variants of the method for such cases will be briefly introduced in the appendix.

## 3.8 GRADIENT CALCULATION

The method for calculating the gradient is given by,

$$\frac{\partial L_{sp}}{\partial c_i} = -\frac{1}{E_{(k,B)}} \frac{\partial E_{(k,B)}}{\partial c_i} = -\frac{1}{P(S_{(k,B)}|c)} \left( \frac{P(b_i = 1, S_{(k,B)}|c)}{c_i} - \frac{P(b_i = 0, S_{(k,B)}|c)}{1 - c_i} \right) \tag{14}$$

where $P(S_{(k,B)}|c) = P(b_i = 1, S_{(k,B)}|c) + P(b_i = 0, S_{(k,B)}|c)$. The backward operation of dynamic programming is conducted using the same computational graph for $P(b_i = 1, S_{(k,B)}|c)$ and $P(b_i = 0, S_{(k,B)}|c)$. Details will be provided in the appendix of the paper.

## 3.9 COMPUTATIONAL COMPLEXITY

The computation of loss and gradient involves both forward and backward dynamic programming processes. The theoretical computational complexity of these operations is quantified as $2 \times O(B \times (2 \times b + 1))$. For the band selection task, our computational complexity is minimal, even lower than the number of floating-point operations required for a $1 \times 1$ convolution. In the future, we plan to apply this method to larger-scale sparse tasks, such as model compression. For models with hundreds of millions of parameters, we aim to explore alternative approximate methods, such as Monte Carlo sampling or hybrid sparsification techniques, to enhance computational efficiency and broaden the applicability of our approach.

## 4 EXPERIMENTS

In our paper, we conducted extensive experiments to validate our spectral band selection methodology using hyperspectral data from various public datasets. Our experimental design included: the comparison with alternative spectral band selection methods, the demonstration of the sparsity effects, the evaluation against other sparsification techniques, and discussions on the hyperparameters. The primary objectives of our experiments were to demonstrate that: 1) our method is highly effective in selecting spectral bands; 2) it achieves notable sparsity; 3) it offers superior capabilities in depicting the relationships between spectral bands. Additional experimental results are presented in the appendix, including analyses of performance across more classifiers, performance variation with different numbers of selected spectral bands, and experiments evaluating performance on other tasks.

### 4.1 DATASETS

In this study, we utilize three distinct and multifaceted datasets for comprehensive analyses and experiments: the KSC hyperspectral dataset Green et al. (1998), the 2013 Houston University dataset (HT2013) GRS (2013) and the 2018 Houston University dataset (HT2018) Prasad et al. (2020). **The KSC dataset**, captured by the AVIRIS sensor over the Kennedy Space Center, Florida, on March 23, 1996, includes a hyperspectral image with 176 of the original 224 spectral bands, after removing 48 bands affected by water vapor noise. This image features a spatial resolution of 18 meters and dimensions of 512×614 pixels, supporting diverse applications such as hyperspectral image classification and mixed pixel decomposition across 13 land cover types. **The HT2013 dataset**, provided by the IEEE GRSS Data Fusion Technical Committee, combines hyperspectral and LiDAR data, including a hyperspectral image with 144 bands covering 380 nm to 1050 nm and a LiDAR-derived Digital Surface Model, both at a spatial resolution of 2.5 meters, and introduces 15 land cover types. **The HT2018 dataset** consists of a hyperspectral image and a LiDAR-derived DSM, both with a spatial resolution of 2.5 meters. It includes 144 spectral bands, ranging from 380 nm to 1050

| Methods | Ground Truth | 5 Selected Bands | | | | | | 10 Selected Bands | | | | | |
| | | SSDGL(2022) | | | DBDA(2020) | | | SSDGL(2022) | | | DBDA(2020) | | |
| | | OA | AA | Kappa | OA | AA | Kappa | OA | AA | Kappa | OA | AA | Kappa |
| All bands | - | 93.9% | 92.1% | 0.929 | 84.2% | 83.0% | 0.837 | 93.9% | 92.1% | 0.929 | 84.2% | 83.0% | 0.837 |
| Cai et al. (2019) | - | 93.1% | 92.0% | 0.921 | 75.9% | 65.8% | 0.728 | 93.7% | 90.2% | 0.929 | 79.9% | 64.2% | 0.771 |
| Li et al. (2021) | - | 93.3% | 91.3% | 0.926 | 74.8% | 71.6% | 0.721 | 93.6% | 91.1% | 0.923 | 75.4% | 70.2% | 0.702 |
| Wu & Yan (2021) | ✓ | 87.9% | 86.8% | 0.862 | 70.0% | 65.1% | 0.677 | 83.3% | 80.8% | 0.810 | 67.5% | 60.0% | 0.574 |
| Li et al. (2023) | - | 91.0% | 88.9% | 0.897 | 72.6% | 66.4% | 0.689 | 93.7% | 90.3% | 0.928 | 74.1% | 70.3% | 0.693 |
| Jia et al. (2023) | ✓ | 93.9% | 91.8% | 0.925 | 83.8% | 82.6% | 0.826 | 93.1% | 91.3% | 0.926 | 81.1% | 79.6% | 0.804 |
| Zhou et al. (2023) | - | 93.9% | 91.6% | 0.932 | 74.8% | 70.1% | 0.714 | 94.2% | 90.9% | 0.934 | 81.4% | 71.0% | 0.788 |
| Yao et al. (2024) | ✓ | 95.3% | 93.9% | 0.954 | 86.2% | 85.0% | 0.854 | 95.5% | 92.6% | 0.950 | 83.3% | 82.5% | 0.805 |
| Ours(CLS) | ✓ | **96.1%** | **94.5%** | **0.959** | **87.4%** | **86.4%** | **0.861** | 95.9% | 93.3% | 0.953 | 84.2% | 83.2% | **0.820** |
| Ours(REC) | - | 94.8% | 92.0% | 0.940 | 77.7% | 68.3% | 0.745 | 94.6% | 91.3% | 0.939 | 83.6% | 78.7% | 0.815 |

Table 1: Comparison of classification accuracy on the KSC dataset using 5 and 10 selected bands.

| Methods | Ground Truth | 5 Selected Bands | | | | | | 10 Selected Bands | | | | | |
| | | SSDGL(2022) | | | DBDA(2020) | | | SSDGL(2022) | | | DBDA(2020) | | |
| | | OA | AA | Kappa | OA | AA | Kappa | OA | AA | Kappa | OA | AA | Kappa |
| All bands | - | 94.7% | 95.2% | 0.943 | 88.5% | 87.7% | 0.876 | 94.7% | 95.2% | 0.943 | 88.5% | 87.7% | 0.876 |
| Cai et al. (2019) | - | 92.9% | 92.4% | 0.924 | 77.5% | 78.1% | 0.757 | 95.1% | 94.8% | 0.948 | 86.3% | 86.6% | 0.853 |
| Li et al. (2021) | - | 92.9% | 92.3% | 0.924 | 67.6% | 68.9% | 0.650 | 94.4% | 94.2% | 0.940 | 75.9% | 77.3% | 0.740 |
| Wu & Yan (2021) | ✓ | 93.4% | 93.0% | 0.929 | 75.9% | 76.0% | 0.740 | 95.3% | 94.4% | 0.950 | 85.6% | 86.6% | 0.844 |
| Li et al. (2023) | - | 93.6% | 92.5% | 0.931 | 79.5% | 72.2% | 0.778 | 95.1% | 93.9% | 0.947 | 85.3% | 86.0% | 0.842 |
| Jia et al. (2023) | ✓ | 94.7% | 94.4% | 0.943 | 77.8% | 78.8% | 0.760 | 96.0% | 95.1% | 0.952 | 83.9% | 83.5% | 0.827 |
| Zhou et al. (2023) | - | 92.4% | 92.0% | 0.916 | 75.9% | 75.5% | 0.748 | 94.9% | 94.3% | 0.941 | 84.3% | 85.0% | 0.836 |
| Yao et al. (2024) | ✓ | 95.1% | 95.1% | 0.945 | 80.5% | 80.0% | 0.788 | 95.8% | 95.6% | 0.954 | 86.3% | 85.1% | 0.852 |
| Ours(CLS) | ✓ | **95.6%** | **95.4%** | **0.952** | **81.0%** | **80.1%** | **0.795** | **96.3%** | **96.2%** | **0.960** | 87.1% | 86.6% | 0.861 |
| Ours(REC) | - | 93.7% | 93.1% | 0.932 | 79.9% | 78.1% | 0.784 | 95.7% | 95.1% | 0.953 | **88.9%** | **88.7%** | **0.880** |

Table 2: Comparison of classification accuracy on the HT2013 dataset using 5 and 10 selected bands.

nm, and covers an area of 349×1905 pixels around the University of Houston campus. The dataset features 15 categories of ground objects, including water bodies, grasslands, trees, buildings, roads, and cars.

## 4.2 Implementation Details

For experiments conducted on public datasets, we followed the partitioning scheme proposed by Li et al. (2020), allocating 5% of the samples for training and band selection, while the remainder were set aside as a test set, aimed at validating the effectiveness of our chosen bands in image classification tasks. The input to the image was a 64x64 patch, the chosen optimizer was Adam, and a batch size of 4 was employed. A single V100 was used for both training and inference. For testing, the raw data was sampled according to the chosen bands, then the corresponding classifier was used for training and testing with a batch size of 16. All other settings remained consistent. In our experiments, we set $\alpha$ to 0.1.The $c_i$ is initially set to 0.5.

## 4.3 Performance of Band Selection on Public Datasets

In this section, we assess the performance of our band selection methodology on various public hyperspectral datasets, and provide a comparative analysis with other state-of-the-art techniques, for example, Yao et al. (2024), Zhou et al. (2023), Jia et al. (2023), Wu & Yan (2021), Li et al. (2021), Li et al. (2023) and Cai et al. (2019). The fewer the selected spectral bands, the greater the challenge for the algorithm. Therefore, following the experimental setup by Zhou et al. (2023), we compare the top 10 and top 5 bands chosen by each method. The network is retrained using the training set data of the selected spectral bands, and the effectiveness of the band selection is evaluated based on the classification performance on its test set. We measure classification accuracy using Overall Accuracy (OA), Average Accuracy (AA), and Kappa coefficient. For this experiment, we employ two deep learning image classification methods SSDGL Zhu et al. (2022) and DBDA Li et al. (2020). The detailed results are presented in Table 1, 2, 4, and 3, where our methods, Ours(CLS) and Ours(REC), are driven by classification and reconstruction tasks, respectively.

Taking into account all experimental indicators, the method proposed in this paper stands out in extracting characteristic spectral bands. This leads to a significant enhancement in both accuracy and efficiency compared to other cutting-edge methods. Our method notably diverges from importance

| Methods | KSC Dataset | | HT2013 Dataset | |
|---|---|---|---|---|
| | 5 Bands | 10 Bands | 5 Bands | 10 Bands |
| Cai et al. (2019) | [55, 63, 70, 85, 86] | [42, 55, 63, 70, 74, 75, 85, 86, 88, 118] | [24, 26, 27, 28, 96] | [24, 26, 27, 28, 31, 42, 96, 98, 136, 142] |
| Li et al. (2021) | [33, 115, 117, 119, 120] | [33, 34, 114, 115, 116, 117, 119, 120, 121, 122] | [64, 65, 114, 142, 143] | [63, 64, 65, 66, 82, 114, 140, 141, 142, 143] |
| Wu & Yan (2021) | [37, 67, 131, 173, 175] | [0, 131, 132, 167, 169, 171, 172, 173, 175, 176] | [49, 50, 52, 55, 143] | [8, 36, 40, 45, 47, 49, 50, 52, 55, 143] |
| Li et al. (2023) | [1, 28, 59, 109, 128] | [1, 28, 59, 84, 88, 96, 109, 128, 134, 175] | [38, 48, 63, 118, 135] | [14, 38, 48, 52, 63, 73, 95, 118, 135, 143] |
| Jia et al. (2023) | [0, 40, 54, 65, 166] | [0, 8, 10, 40, 54, 55, 65, 95, 143, 166] | [27, 64, 65, 89, 107] | [22, 25, 26, 27, 64, 65, 89, 105, 107, 108] |
| Zhou et al. (2023) | [32, 73, 125, 170, 174] | [0, 32, 73, 125, 167, 169, 170, 171, 172, 174] | [0, 1, 121, 122, 123] | [0, 1, 2, 3, 120, 121, 122, 123, 124, 125] |
| Yao et al. (2024) | [13, 34, 55, 76, 93] | [13, 30, 34, 43, 55, 76, 91, 93, 105, 117] | [6, 29, 51, 67, 90] | [6, 28, 29, 50, 51, 67, 79, 90, 109, 137] |
| Ours(CLS) | [25, 26, 28, 29, 96] | [24, 25, 26, 30, 31, 32, 37, 95, 96, 118] | [4, 6, 8, 68, 142] | [3, 4, 5, 36, 63, 64, 73, 87, 88, 143] |
| Ours(REC) | [13, 32, 35, 37, 173] | [13, 17, 19, 21, 26, 32, 37, 63, 96, 120, 173] | [12, 80, 81, 130, 131] | [11, 12, 19, 46, 73, 80, 81, 82, 130, 131] |

Table 3: Detailed selected bands for each method across the KSC and HT2013 datasets, organized by the number of selected bands, sorted in ascending order.

| Methods | Ground Truth | 5 Selected Bands | | | | | | 10 Selected Bands | | | | | |
|---|---|---|---|---|---|---|---|---|---|---|---|---|---|
| | | SSDGL(2022) | | | DBDA(2020) | | | SSDGL(2022) | | | DBDA(2020) | | |
| | | OA | AA | Kappa | OA | AA | Kappa | OA | AA | Kappa | OA | AA | Kappa |
| All bands | - | 98.0% | 95.4% | 0.975 | 89.4% | 85.5% | 0.862 | 98.0% | 95.4% | 0.975 | 89.4% | 85.5% | 0.862 |
| Li et al. (2021) | - | 96.2% | 89.7% | 0.950 | 80.4% | 70.8% | 0.770 | 96.9% | 91.2% | 0.956 | 83.7% | 77.6% | 0.798 |
| Wu & Yan (2021) | ✓ | 96.0% | 91.6% | 0.948 | 82.3% | 72.2% | 0.771 | 96.4% | 89.3% | 0.953 | 84.9% | 79.7% | 0.803 |
| Li et al. (2023) | - | 95.9% | 88.4% | 0.943 | 79.2% | 68.9% | 0.760 | 97.1% | 90.8% | 0.957 | 83.9% | 78.5% | 0.805 |
| Jia et al. (2023) | ✓ | 96.8% | 92.2% | 0.943 | 77.2% | 62.5% | 0.699 | 97.4% | 93.7% | 0.966 | 85.1% | 76.4% | 0.807 |
| Zhou et al. (2023) | - | 96.7% | 91.0% | 0.952 | 82.5% | 75.2% | 0.780 | 97.3% | 92.6% | 0.959 | 84.2% | 79.4% | 0.814 |
| Yao et al. (2024) | ✓ | 97.5% | 93.5% | 0.965 | 85.2% | 78.0% | 0.805 | 97.9% | 93.1% | 0.970 | 87.0% | 82.0% | 0.835 |
| Ours(CLS) | ✓ | **98.2%** | **94.3%** | **0.977** | **85.6%** | **80.0%** | **0.811** | **98.3%** | **94.9%** | **0.978** | **88.1%** | **84.4%** | **0.845** |
| Ours(REC) | - | 97.3% | 92.5% | 0.958 | 84.9% | 79.5% | 0.805 | 97.9% | 93.3% | 0.970 | 86.5% | 80.5% | 0.828 |

Table 4: Comparison of classification accuracy on the HT2018 dataset using 5 and 10 selected bands.

ranking approaches because it is associated with the number of spectral segments selected. The optimal results obtained with 10 spectral segments are not necessarily equivalent to those with 5, an intuitively reasonable notion. Furthermore, there are cases where choosing 5 spectral segments results in greater accuracy than opting for 10, indicating that a higher number of spectral segments doesn't always correspond with better classification performance. This is in line with the insights provided by Morales et al. (2021), who highlight that an increase in spectral bands can lead to excessive redundancy and may not improve classification efficacy. The high-dimensional nature of data further complicates model convergence and generalization. The current experimental outcomes also adhere to the Hughes phenomenon, whereby the classification accuracy first improves and then diminishes as the number of spectral bands grows.

### 4.4 SPARSITY OF IMPORTANCE LEVELS

Fig. 3 presents the evolution of importance levels for different bands throughout the training process. Primarily, these can be grouped into two categories: selected bands and non-selected ones. The importance levels for the selected bands display an initial decrease followed by a subsequent increase, while those for the non-selected bands show a consistent decline. the turning point of this decline corresponds to the saddle point identified in Theorem 2 of Section 3.4, indicative of typical behavior in the optimization process. Furthermore, the band sparsification process exhibits a feature of sparsity in sequence. We infer that, during the learning process, bands with significant importance are sparsified first, which subsequently influences the model's evaluation of the importance of later bands, thereby creating a sequential sparsification trend.

### 4.5 COMPARISON WITH OTHER SPARSITY STRATEGIES

We will compare our sparsity technique with other commonly used sparsity strategies. Our comparison encompasses two aspects: the degree of sparsity and the classification performance of selected bands. Fig. 2 compares the distribution of band importance levels after 300 training epochs for various sparsity losses such as L1 and L2 norm. As shown in the figure, our proposed method exhibits a superior degree of sparsity. Furthermore, Table 5 presents the selected bands and their respective classification results. The Gumbel-Sigmoid method, proposed by Zhou et al. (2021), achieves enforced sparsity by continuously tightening the temperature coefficient $T$. In contrast, our method is not constrained by such a parameter, offering greater flexibility The loss function proposed in this

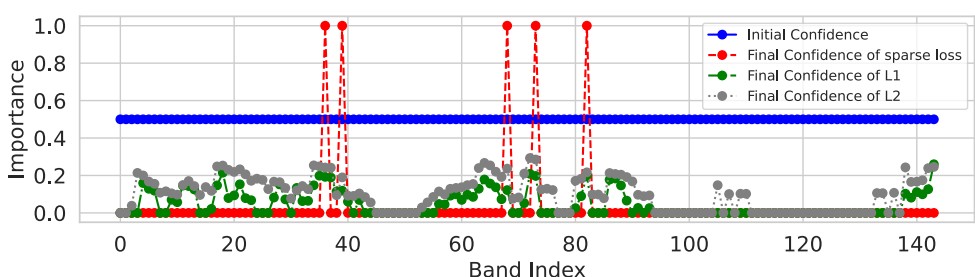

Figure 2: Sparsity of importance levels compared with other sparse loss.

| Methods | Selected Bands | SSDGL | | | DBDA | | |
|---|---|---|---|---|---|---|---|
| | | OA | AA | Kappa | OA | AA | Kappa |
| L1 | [143, 18, 72, 82, 35] | 94.5% | 94.2% | 0.940 | 78.6% | 79.3% | 0.769 |
| L2 | [72, 73, 64, 34, 65] | 94.3% | 94.0% | 0.938 | 76.4% | 76.4% | 0.745 |
| Gumbel | [36, 66, 82, 63, 92] | 94.7% | 94.8% | 0.941 | 80.4% | 79.4% | 0.789 |
| $L_{sp}$ | [4, 6, 8, 68, 142] | **95.6%** | **95.4%** | **0.952** | **81.0%** | **80.1%** | **0.795** |

Table 5: Comparison of classification accuracy when selecting 5 bands on the HT2013 dataset using different sparsification losses.

paper allows for the learning of more representative bands and achieves superior performance in the classification task.

### 4.6 Exploring the Effectiveness of $L_{sp}$ in Mining Relationships

In this experiment, we established a scenario containing $K$ ($K \in \{50, 100\}$) spectral bands ($c$, where $c_i \in [0, 1]$), and constructed a $K \times K$ binary weight matrix ($A$, where $a_{i,j} > 0$) to describe the relationships among these bands. The goal was to maximize the sum of binary weights by selecting 5 and 10 bands ($|cA * c|_1$). To achieve this, we applied four different methods: L1, L2, Gumbel, and EM, each run 40 times under various random seed settings to select the band combination that maximized the weight sum. By comparing the results generated by these methods, we utilized statistical tests (T-tests) to assess and demonstrate the superiority of the EM method in mining and utilizing the relationships between spectral bands. The experiment results indicated that the EM method showed a statistically significant advantage ($p < 0.05$) in depicting the relationships between bands compared to the L1, L2, and Gumbel methods, thereby proving its potential and effectiveness in solving such problems (see Table 6).

### 4.7 Hyperparameter Analysis

In this section, we delve into the determination of optimal hyperparameters for the sparsity loss function, mainly examining the degree of sparsity and classification accuracy. Through our experiments, we found that a hyperparameter of 0.05 produced the best results. As shown in Table 7, when the

| Comparison | 5 from 50 bands | | 10 from 50 bands | | 5 from 100 bands | | 10 from 100 bands | |
|---|---|---|---|---|---|---|---|---|
| | t-value | p-value | t-value | p-value | t-value | p-value | t-value | p-value |
| $L_{sp}$ vs L1 | 5.224 | $< 0.001$ | 4.989 | $< 0.001$ | 4.732 | $< 0.001$ | 7.658 | $< 0.001$ |
| $L_{sp}$ vs L2 | 12.048 | $< 0.001$ | 9.930 | $< 0.001$ | 4.425 | $< 0.001$ | 7.727 | $< 0.001$ |
| $L_{sp}$ vs Gumbel | 9.173 | $< 0.001$ | 6.543 | $< 0.001$ | 2.207 | 0.036 | 13.549 | $< 0.001$ |

Table 6: Statistical comparison using different sparsification losses for selecting 5 and 10 spectral bands from 50 and 100 bands.

| $\alpha$ | Selected Bands | Sparsity Status | OA |
|---|---|---|---|
| 0.1 | [25, 26, 28, 29, 96] | True | 96.1% |
| 0.05 | [42, 43, 95, 96, 118] | True | **96.4%** |
| 0.03 | [42, 43, 95, 96, 118] | True | **96.4%** |
| 0.02 | [40, 41, 42, 43, 96] | False | 95.6% |

Table 7: Classification accuracy on the KSC dataset using 5 selected bands with varying hyperparameters of sparsity loss.

hyperparameter is below 0.02, the parameters of the band importance fail to converge and to achieve sparsity (see Fig. 4). While the bands selected at 0.05 and 0.03 are the same, the convergence speed is faster at 0.05. Regarding the performance of the sparsity loss hyperparameter in image classification tasks, it shows an initial increase followed by a decrease as the hyperparameter decreases. We believe the reason for this trend is that during the band selection process, the classification gradient and the sparsity gradient are balanced and antagonized. The larger the hyperparameter for sparsity loss, the stronger the sparsity gradient and the most important information transmitted by the classification gradient is more likely to be masked. As the sparsity gradient is minimized, a sufficient interplay between the two is allowed, hence selecting the most representative bands.

However, when the hyperparameter for sparsity loss is less than 0.03, it means that the sparsity gradient is not strong enough to compete with the classification gradient to make the parameters as sparse as possible. After all, the classification task hopes to use as many features as possible, while the sparsity task hopes to ignore as many bands as possible, creating a certain degree of contradiction. In addition, the decline in classification accuracy at 0.02 further confirms our assumption, i.e., importance sparsity helps alleviate issues brought about by inaccurate importance levels, thus learning more representative bands.

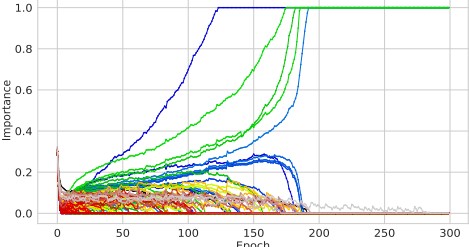

Figure 3: Changes in spectral band importance over training epochs. As illustrated, the sparsification process exhibits certain sequential characteristics, that is, bands selected earlier may influence the selection of subsequent bands.

Figure 4: Sparsity process of 0.002. At 0.02, the gradient brought about by the sparsity loss is insufficient to counterbalance the classification gradient, thus creating a watershed at 0.02 that makes it difficult for the model to be sparse.

## 5 CONCLUSION

This research has made significant contributions to overcoming the limitations of existing band selection methods in hyperspectral imaging technologies. We have proposed a novel deep-learning band selection method based on importance sparsity to address the issue of depicting the relationships between spectral bands. The introduction of Sparsity Loss has markedly improved band selection performance and convergence. Our method's validation on public datasets demonstrates its robustness and practical applicability. In the future, we aim to further investigate the relationship between the representation of importance in band selection, subsequent models, and input data. Additionally, we plan to explore the effectiveness of our proposed sparsification method in other diverse fields.

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

APPENDIX

The appendix includes the following sections:

1. Disclosure of the code used in the research;

2. Detailed proofs of the two theorems presented in Section 3.4;

3. The application of dynamic programming for solving $P(b_i = 1, S_{(k,B)}|c)$ and $P(b_i = 0, S_{(k,B)}|c)$, along with the computation of gradients;

4. Additional experimental results.

5. Analysis and Interpretation of Sparse Loss from an another perspective;

6. Design considerations regarding $c_i$ being within the interval [0,1];

7. Variants of the method under a locally uniform distribution;

8. Additional Thoughts on Sparsification and Spectral Band Selection.

## A  CODE DISCLOSURE

Our code is currently available at https://anonymous.4open.science/r/Sparse-Hyperspectral-Band-Selection-Based-on-Expectation-Maximization-4EEC and will be made public on GitHub after the article is published. Should the link fail to open, the code has been provided in a zip file within the Supplementary Material.

## B  PROOFS OF THEOREMS IN SECTION 3.4

**Theorem 1:** Within the range of $c_i \in [0, 1]$, $E_{(k,B)}$ assumes values in $[0, 1]$. It achieves a value of 1 if and only if, within the set $\{c_i\}$, $k$ elements are equal to 1 and $(B - k)$ elements are equal to 0.

*Proof:*

$$\because E_{(k,B)} = \sum_{\pi \in C_{(k,B)}} P(S_{(k,B)}|\pi, c)$$

$$\because P(S_{(k,B)}|\pi, c) \geq 0$$

$$\therefore E_{(k,B)} \geq 0$$

$$\because \sum_{i=0}^{B} E_{(i,B)} = \prod_{j=0}^{B} [P(b_j = 0|C) + P(b_j = 1|C)] = 1$$

$$\therefore E_{(k,B)} \leq \sum_{i=0}^{B} E_{(i,B)} = 1$$

$$\therefore \sum_{i=0}^{B} E_{(i,B)} - E_{(k,B)} = \sum_{i=0,i\neq k}^{B} E_{(i,B)}$$

When $E_{(k,B)} = 1$, then $\sum_{i=0,i\neq k}^{B} E_{(i,B)} = 0$, it follows that in the set $\{c_i\}$, there are $k$ elements equal to 1 and the rest are 0.

**Theorem 2:** Within $c_i \in (0, 1)$, $E_{(k,B)}$ has no local maxima but only a saddle point at $c_1 = c_2 = \ldots = c_B = \frac{k}{B}$.

*Proof:*

$$\because E_{(k,B)} = \sum_{\pi \in C_{(k,B)}} P(S_{(k,B)}|\pi, c)$$

$$\therefore E_{(k,B)} = c_i E_{(k-1,B-1)} + (1 - c_i) E_{(k,B-1)}$$

$$\therefore E_{(k,B)} = (c_i(1 - c_j) + c_j(1 - c_i)) E_{(k-1,B-2)} + c_i c_j E_{(k-2,B-2)}$$

$$\therefore \frac{\partial E_{(k,B)}}{\partial c_i} = (1 - 2c_j) E_{(k-1,B-2)} + c_j E_{(k-2,B-2)}$$

$$\therefore \frac{\partial E_{(k,B)}}{\partial c_i} - \frac{\partial E_{(k,B)}}{\partial c_j} = 2(c_i - c_j) E_{(k-1,B-2)} - (c_i - c_j) E_{(k-2,B-2)}$$

$$\therefore \frac{\partial E_{(k,B)}}{\partial c_i} - \frac{\partial E_{(k,B)}}{\partial c_j} = (c_i - c_j)(2E_{(k-1,B-2)} - E_{(k-2,B-2)})$$

Substituting $2E_{(k-1,B-2)} = E_{(k-2,B-2)}$ into $\frac{\partial E_{(k,B)}}{\partial c_i} = 0$, we get $E_{(k-1,B-2)} = E_{(k-2,B-2)} = 0$, which are discarded.

$$\because c_i = c_j$$

$$\therefore c_1 = c_2 = ... = c_B$$

Substituting into $\frac{\partial E_{(k,B)}}{\partial c_i} = 0$, we obtain,

$$C_{B-1}^{k-1} c_i^{k-1} (1 - c_i)^{B-k} - C_{B-1}^{k} c_i^{k} (1 - c_i)^{B-1-k} = 0$$

$$\therefore c_i = \frac{k}{B}$$

$$\frac{\partial E_{(k,B)}}{\partial c_i \partial c_i} = 0$$

$$\frac{\partial E_{(k,B)}}{\partial c_i \partial c_j} = \frac{1}{B-1} - \frac{2}{k-1} - 2E_{(k-1,B-2)} + E_{(k-2,B-2)}$$

Substituting $c_1 = c_2 = ... = c_B = \frac{k}{B}$, we get,

$$\frac{\partial E_{(k,B)}}{\partial c_i \partial c_j} = \frac{1}{B-1} - \left(\frac{2}{k-1} + \frac{1}{B-k}\right) \frac{k}{B} = \delta$$

$$\therefore H = \begin{pmatrix} 0 & \delta & \cdots & \delta \\ \delta & 0 & \cdots & \delta \\ \vdots & \vdots & \ddots & \vdots \\ \delta & \delta & \cdots & 0 \end{pmatrix}$$

$$\therefore eig(H) = eig(\delta(J - I))$$

$$\because J = \begin{pmatrix} 1 & 1 & \cdots & 1 \\ 1 & 1 & \cdots & 1 \\ \vdots & \vdots & \ddots & \vdots \\ 1 & 1 & \cdots & 1 \end{pmatrix}$$

The matrix $J$ has eigenvalues $[B, 0, \ldots, 0, 0]$. Therefore, the eigenvalues of $H$ are $[\delta(B - 1), -\delta, \ldots, -\delta, -\delta]$. Given that $\delta(B - 1)$ and $-\delta$ have opposite signs, it follows that the solution $c_0 = c_1 = \ldots = c_B = \frac{k}{B}$ represents a saddle point.

## C  GRADIENT CALCULATION USING DYNAMIC PROGRAMMING

The recursive formula for the backward algorithm is defined as follows,

$$
z_j^i = \begin{cases}
p(b_B = 0|c) & \text{if } j = 2k+1 \text{ and } i = B, \\
z_j^{i+1} p(b_i = 0|c) & \text{if } j = 2k+1 \text{ and } i \neq B, \\
p(b_B = 1|c) & \text{if } j = 2k \text{ and } i = B, \\
z_{j+1}^{i+1} p(b_i = 1|c) & \text{if } j = 2k \text{ and } i \neq B, \\
(z_j^{i+1} + z_{j+1}^{i+1}) p(b_i = 0|c) & \text{if } j = 1, 3, \ldots, 2k-1 \text{ and } i \neq B, \\
(z_{j+1}^{i+1} + z_{j+2}^{i+1}) p(b_i = 1|c) & \text{if } j = 2, 4, \ldots, 2k-2 \text{ and } i \neq B, \\
0 & \text{else .}
\end{cases}
\tag{15}
$$

Combine the forward algorithm and backward algorithm to calculate $P(b_i = 1, S_{(k,B)}|c)$ and $P(b_i = 0, S_{(k,B)}|c)$,

$$
q_j^i = \begin{cases}
\frac{a_j^i z_j^i}{p(b_i=0|c)} & \text{if } j = 1, 3, \ldots, 2k+1 \\
\frac{a_j^i z_j^i}{p(b_i=1|c)} & \text{if } j = 2, 4, \ldots, 2k.
\end{cases}
\tag{16}
$$

The term $q_j^i$ represents the sum of probabilities for all selections passing through node $(i, j)$. Since $a_j^i$ and $z_j^i$ involve multiplying $p(b_i = 0|c)$ or $p(b_i = 1|c)$ twice, the product needs to be divided by one. Therefore, the method to calculate $P(b_i = 1, S_{(k,B)}|c)$ and $P(b_i = 0, S_{(k,B)}|c)$ is,

$$
P(b_i = 0, S_{(k,B)}|c) = \sum_{j=1}^{k+1} q_{2j-1}^i,
\tag{17}
$$

$$
P(b_i = 1, S_{(k,B)}|c) = \sum_{j=1}^{k} q_{2j}^i.
\tag{18}
$$

Therefore, the gradient of the sparse loss can be represented as,

$$
\frac{\partial L_{sp}}{\partial c_i} = -\frac{1}{E_{(k,B)}} \frac{\partial E_{(k,B)}}{\partial c_i}
\tag{19}
$$

$$
= -\frac{1}{P(S_{(k,B)}|c)} \left( \frac{P(b_i = 1, S_{(k,B)}|c)}{c_i} - \frac{P(b_i = 0, S_{(k,B)}|c)}{1 - c_i} \right)
\tag{20}
$$

$$
= -\frac{1}{a_{2k}^B + a_{2k+1}^B} \left( \frac{\sum_{j=1}^{k} q_{2j}^i}{c_i} - \frac{\sum_{j=1}^{k+1} q_{2j-1}^i}{1 - c_i} \right)
\tag{21}
$$

$$
\tag{22}
$$

## D  ADDITIONAL EXPERIMENTAL RESULTS.

In this section, we performed additional experimental analyses, focusing on three main aspects: the performance of the experimental results when tested with an alternative classification network, the accuracy variation across different spectral band selections, and the performance of the band selection method in other tasks.

### D.1  EXPERIMENTS ON DIFFERENT CLASSIFIERS

To further evaluate performance across a broader range of classifiers, we selected two additional classifiers (Two-CNN Yang et al. (2017) and CDSFT Qiu et al. (2023)) and compared the results across four classifiers in total. Table 8 presents the experimental results.

### D.2  ACCURACY VARIATION IN DIFFERENT SPECTRAL BANDS

In our experiments, we observed that fewer spectral bands often led to higher accuracy. While this aligns with the Hughes phenomenon, the underlying principles and explanations specific to neural

| Methods | Ground Truth | Two-CNN(2018) | | | DBDA(2020) | | | SSDGL(2022) | | | CDSFT(2023) | | |
|---------|:---:|------|------|-------|------|------|-------|------|------|-------|------|------|-------|
| | | OA | AA | Kappa | OA | AA | Kappa | OA | AA | Kappa | OA | AA | Kappa |
| All bands | ✓ | 81.2% | 75.0% | 0.785 | 84.2% | 83.0% | 0.837 | 93.9% | 92.1% | 0.929 | 98.2% | 97.2% | 0.978 |
| Wu & Yan (2021) | ✓ | 74.7% | 64.3% | 0.711 | 70.0% | 65.1% | 0.677 | 87.9% | 86.8% | 0.862 | 93.4% | 90.0% | 0.925 |
| Jia et al. (2023) | ✓ | 77.2% | 71.5% | 0.758 | 83.8% | 82.6% | 0.826 | 93.9% | 91.8% | 0.925 | 96.0% | 93.1% | 0.955 |
| Ours(CLS) | ✓ | **82.2%** | **74.8%** | **0.792** | **87.4%** | **86.4%** | **0.861** | **96.1%** | **94.5%** | **0.959** | **98.0%** | **95.6%** | **0.968** |

Table 8: Comparison of classification accuracy on the KSC dataset using different classifiers.

| | 3 bands | 4 bands | 5 bands | 7 bands | 10 bands | 20 bands | All Bands |
|---|---------|---------|---------|---------|----------|----------|-----------|
| OA | 91.0% | 95.1% | 96.1% | **96.5%** | 95.9% | 94.8% | 93.9% |
| AA | 90.7% | 92.2% | **94.5%** | 94.4% | 93.3% | 92.7% | 92.1% |
| Kappa | 0.895 | 0.938% | **0.959** | 0.956 | 0.953 | 0.941 | 0.929 |

Table 9: Accuracy variation under SSDGL in different spectral bands.

networks are not entirely clear. We found this effect to be influenced by the experimental dataset and the classification methods used for validation, with some methods on certain datasets showing a particularly pronounced effect (e.g., accuracy with 5 bands > 10 bands > full spectrum). We suspect this may be due to the increased number of spectral bands without a corresponding increase in data volume, which makes the model more prone to overfitting during optimization, thus impacting generalization. Table 9 provides additional experimental results that illustrate these accuracy changes.

## D.3 EXPERIMENTS IN DIFFERENT TASKS

We included anomaly detection and target detection experiments in Tables 10 and 11. Target detection was conducted on the Viareggio 2013 dataset Acito et al. (2016) using the ACDA Hu et al. (2021) testing method, while anomaly detection was performed on the San Diego II dataset with the testing method HTD-IRN Shen et al. (2023).

| Methods | AUC |
|---------|-----|
| All bands | 0.801 |
| Cai et al. (2019) | 0.808 |
| Li et al. (2021) | 0.757 |
| Zhou et al. (2023) | 0.821 |
| Ours(REC) | **0.839** |

Table 10: Comparison of AUC on the Viareggio 2013 dataset (anomaly detection).

| Methods | AUC |
|---------|-----|
| All bands | 0.998 |
| Cai et al. (2019) | 0.706 |
| Li et al. (2021) | 0.864 |
| Zhou et al. (2023) | 0.753 |
| Ours(REC) | **0.914** |

Table 11: Comparison of AUC on the San Diego II dataset (target detection).

## E ANALYSIS AND INTERPRETATION OF SPARSE LOSS

In order to explore the operational principles behind the sparsity loss function $L_{sp}$, we derive the following insights: *The sparsity loss $L_{sp}$ can be conceptualized as a two-step process. Initially, each band is assigned a pseudo label based on its context. Following this, the pseudo label and its associated confidence level are subjected to cross-entropy training. This method aims to train the importance of bands to achieve maximal sparsity.*

To prove this, we first derive its gradient,

$$\frac{\partial L_{sp}}{\partial c_i} = -\frac{1}{P(S_{(k,B)}|c)}\left(\frac{P(b_i = 1, S_{(k,B)}|c)}{c_i} - \frac{P(b_i = 0, S_{(k,B)}|c)}{1 - c_i}\right) \tag{23}$$

$$= -\frac{P(b_i = 1, S_{(k,B)}|c)}{P(S_{(k,B)}|c)}\frac{1}{c_i} + \frac{P(b_i = 0, S_{(k,B)}|c)}{P(S_{(k,B)}|c)}\frac{1}{1 - c_i}. \tag{24}$$

In the equation,

$$\frac{P(b_i = 1, S_{(k,B)}|c)}{P(S_{(k,B)}|c)} + \frac{P(b_i = 0, S_{(k,B)}|c)}{P(S_{(k,B)}|c)} = 1. \tag{25}$$

Let,

$$\hat{p}_i = \frac{P(b_i = 1, S_{(k,B)}|c)}{P(S_{(k,B)}|c)}, \tag{26}$$

$$1 - \hat{p}_i = \frac{P(b_i = 0, S_{(k,B)}|c)}{P(S_{(k,B)}|c)}, \tag{27}$$

we obtain,

$$\frac{\partial L_{sp}}{\partial c_i} = -\frac{\hat{p}_i}{c_i} + \frac{1 - \hat{p}_i}{1 - c_i} \tag{28}$$

$$= -\frac{\partial \hat{p}_i \log c_i}{\partial c_i} - \frac{\partial (1 - \hat{p}_i) \log(1 - c_i)}{\partial c_i} \tag{29}$$

$$= -\frac{\partial (\hat{p}_i \log c_i + (1 - \hat{p}_i) \log(1 - c_i))}{\partial c_i} \tag{30}$$

where $\hat{p}_i$ can be viewed as a constant. It is an estimate of the probability for selecting the $i$th band. This gradient is equivalent to the cross-entropy gradient of the label $\hat{p}_i$ and probability value $c_i$,

$$\arg\min_C L_{sp} = \arg\min_C - \sum_{i=1}^{B} (\hat{p}_i \log c_i + (1 - \hat{p}_i) \log(1 - c_i)). \tag{31}$$

In this equation, $\hat{p}_i$ can be interpreted as the pseudo labels. These pseudo labels are calculated as the ratio of the sum of the probabilities of all selections passing through the filled (or hollow) nodes to the sum of the probabilities of all selections. This aids in understanding the operational principle of Sparse Loss. We also analyze its convergence process in the experimental section. The original pseudocode is outlined in Algorithm 1, and the version using pseudo labels is presented in Algorithm 2. These two are equivalent in function.

---

**Algorithm 1** Sparse Loss Based on Expectation-Maximization Algorithm

---

1: Initialize band selection weights $C^{(0)}$ with 0.5, maximum training epochs $T$,
2: $t = 0$
3: **repeat**
4:     **E Step**: Calculating probability $p(b_i = 1|c)$ and $p(b_i = 0|c)$ based on importance $c$, and $L_{sp}$ can be obtained by dynamic programming.
5:     **M Step**: Updating band selection weights $C$ with $L_{sp}$ and $L_{task}$,

$$c_i^{(t+1)} = c_i^{(t)} - \nabla(L_{task} + \alpha L_{sp})$$

6:     $t = t + 1$
7: **until** $t > T$

---

---

**Algorithm 2** Sparse Loss in a Pseudo Label Version

---

1: Initialize band selection weights $C^{(0)}$ with 0.5, maximum training epochs $T$,
2: $t = 0$
3: **repeat**
4:     **Pseudo Label Calculation**: Calculating $\hat{p}_i$ according to Equation 26 and 27 based on $p(b_i = 1|c)$ and $p(b_i = 0|c)$ for each band.
5:     **Cross Entropy Training**: Updating band selection weights $C$ with $L_{sp}$ and $L_{task}$,

$$c_i^{(t+1)} = c_i^{(t)} - \nabla(L_{task} - \alpha \sum_{i=1}^{B}(\hat{p}_i \log c_i + (1 - \hat{p}_i)\log(1 - c_i)))$$

6:     $t = t + 1$
7: **until** $t > T$

---

## F    MODEL DESIGN CONCERNING $c_i$ WITHIN [0,1]

To ensure that $c_i$ remains within the range [0,1] during the training process, we adopted the following strategy,

$$c_i = \min\{\max\{w_i, 0\}, 1\}. \tag{32}$$

The weight $W = \{w_0, w_1, \ldots, w_B\}$ associated with each band reflects its likelihood of selection.

**Comparison of Normalization Methods:** This part compares various normalization techniques, with our normalization method showing notable convergence speed advantages. Our method maintains classification accuracy despite faster speeds, leading to more distinct band selection and better accuracy. Fig. 6 contrasts the convergence of the sigmoid function and our method, underscoring the struggle of other methods to converge. We assess accuracies with 30 spectral bands (TABLE 5), chosen to ensure fairness, as non-linear methods falter at higher sparsity levels (e.g., 5 or 10 bands). Our method demonstrates a clear advantage in these experiments.

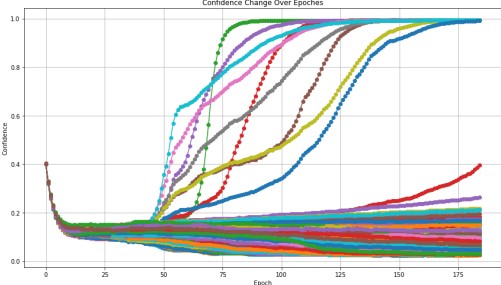

| Methods | OA | AA | Kappa |
|---|---|---|---|
| Exp | 92.5% | 91.2% | 0.916 |
| Softmax | 92.0% | 90.9% | 0.914 |
| Sigmoid | 94.0% | 92.1% | 0.919 |
| Ours | **94.2%** | **92.6%** | **0.930** |

Figure 5: Classification accuracy on the KSC dataset using 30 selected bands with various normalization methods.

Figure 6: Convergence speed with sigmoid. It is still converging slowly at 200 epochs, and its rate of change is significantly lower than our method. The convergence speed of our method is shown in Fig. 3.

## G    VARIANTS UNDER A LOCALLY UNIFORM DISTRIBUTION

In certain specific application scenarios, the assumption of a uniform distribution may not be applicable. Practical needs may require selecting a specific number of bands within different spectral ranges. For instance, for color imaging, we might need to choose several bands in the red, green, and blue spectral regions, while for nighttime imaging, bands in the near-infrared region are more relevant. These practical considerations are important and must be addressed.

As long as a local uniform distribution is maintained, our method remains applicable. In the example above, a specific spectral range, such as the near-infrared region, follows a uniform distribution within that region. Each local region can use dynamic programming to compute the path probability sum

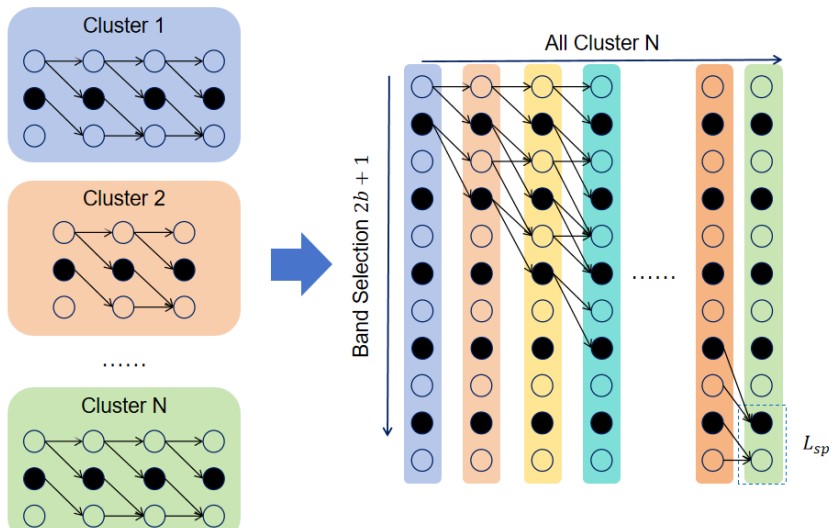

Figure 7: Schematic diagram of the method extension under the local uniform distribution prior.

within the region, and the probability sums from all regions can then be combined to calculate the overall probability (see Figure 7). This does not increase computational complexity. Assuming one band is selected from each spectral region, with $N$ spectral regions and $b$ bands to be selected, the computational complexity is $O(B \times 3 + N \times (2 \times b + 1))$. Since each spectral region contains at least two bands, $2 \times N \leq B$, the complexity is $O(B \times 3 + N \times (2 \times b + 1)) \leq O(B \times 3 + B \times (b + 0.5)) = O(B \times (b + 3.5))$, which is on the same order of magnitude as the original algorithm and, in most cases, even smaller.

For clustering methods, which aim to reduce redundancy by grouping bands into clusters, selecting one band from each cluster also involves modeling the relationships between bands and addressing inaccuracies in band confidence. The method shown in Figure 7 can effectively solve these problems, and this will be a focus of our future research.

## H  ADDITIONAL THOUGHTS ON SPARSIFICATION AND SPECTRAL BAND SELECTION

### H.1  THEOREM 3

*Let $\mathbf{f}$ be a function representing a neural network with $L$ layers, each employing the Rectified Linear Unit (ReLU) activation function. Let $\mathbf{x} \in \mathbb{R}^B$ be the input vector to the network, where each element $x_i$ corresponds to a spectral band. Then, under the assumption that each layer of the network performs a linear transformation followed by the ReLU activation, the output of the network, $\mathbf{f}(\mathbf{x})$, can be represented as a piecewise linear function of the input vector $\mathbf{x}$. Specifically, there exists a set of coefficients $\{v_i\}$ and biases $\{a_i\}$ such that the network output is given by:*

$$\mathbf{f}(\mathbf{x}) = \sum_{i=1}^{B} v_i x_i + a_i, \tag{33}$$

*where the sum is over the spectral bands indexed by $i$.*

### H.2  PROOF OF THEOREM 3

Based on the derivation by Hein et al. Hein et al. (2019), taking a fully connected layer as an example, the output features of each layer of the neural network can be expressed as follows:

$$f^{(k)}(x) = W^{(k)} g^{(k-1)}(x) + b^{(k)}, \tag{34}$$

$$g^{(k)}(x) = \sigma(f^{(k)}(x)), \quad k = 1, \ldots, L. \tag{35}$$

Here, $g^{(k)}$ epresents the output of the $k$-th layer, with $g^{(0)}(x) = x$ denoting the input data. $f^{(L+1)}(x) = W^{(L+1)}g^{(L)}(x) + b^{(L+1)}$ represents the final classifier output. $W^{(k)}$ and $b^{(k)}$ are the weight and bias of the $k$-th layer, respectively, and $\sigma(x)$ represents the ReLU activation function.

Define diagonal matrices $\Delta^{(l)}, \Sigma^{(l)} \in \mathbb{R}^{n_l \times n_l}, \quad l = 1, \ldots, L$ to represent the ReLU function,

$$\Sigma^{(l)}(x)_{ij} = \begin{cases} 1 & \text{if } i = j \text{ and } f_i^{(l)}(x) > 0, \\ 0 & \text{else.} \end{cases} \tag{36}$$

The expression for $f^{(k)}(x)$ can then be represented as,

$$f^{(k)}(x) = W^{(k)} \Sigma^{(k-1)}(x) \left( W^{(k-1)} \Sigma^{(k-2)}(x) \right.$$
$$\left. \times \left( \ldots \left( W^{(1)}x + b^{(1)} \right) \ldots \right) + b^{(k-1)} \right) + b^{(k)}. \tag{37}$$

Combining these, we can obtain the final linear function,

$$f^{(k)}(x) = V^{(k)}x + a^{(k)}, \tag{38}$$

where $V^{(k)}$ and $a^{(k)}$ can be respectively expressed as,

$$V^{(k)} = W^{(k)} \left( \prod_{l=1}^{k-1} \Sigma^{(k-l)}(x) W^{(k-l)} \right), \tag{39}$$

$$a^{(k)} = b^{(k)} + \sum_{l=1}^{k-1} \left( \prod_{m=1}^{k-l} W^{(k+1-m)} \Sigma^{(k-m)}(x) \right) b^{(l)}. \tag{40}$$

The final classifier output $f^{(L+1)}$ can be viewed in the following form,

$$f^{(L+1)} = \sum_{i=1}^{B} v_i^{(L+1)} x_i + a^{(L+1)}, \tag{41}$$

where $x_i$ represents the input value of the $i$-th spectral band, and $V^{(L+1)} = [v_1^{(L+1)}, v_2^{(L+1)}, \ldots, v_B^{(L+1)}]$. From this, the output of the neural network can be seen as a linear combination of the values across different spectral bands, thus proving Theorem 1.

## H.3 THEOREM 4

*Let $S_k$ be the subset of spectral bands selected based on the highest importance scores, and let $T_k$ be the subset of the actual top $k$ most important bands. We assert that $S_k = T_k$ if the selection mechanism for importance scores satisfies the following condition: for any spectral band $i$ not in $T_k$, and any spectral band $j$ in $T_k$, the importance scores are respectively 0 and 1. In other words, the sparse importance score preserves the ordering of the actual importance scores for the top $k$ bands.*

## H.4 PROOF OF THEOREM 4

First, we need to define the method for calculating the actual contribution of different spectral bands in a classification task. We can consider the actual contribution as a distance metric function related to $v_i^{(L+1)} x_i$, such as $L2$, $L1$, and others.

Given the $L2$ norm, the actual importance of spectral band $i$, denoted by $I_i$, can be expressed as,

$$I_i = \left\| v_i^{(L+1)} x_i \right\| = x_i \left\| v_i^{(L+1)} \right\|. \tag{42}$$

Consider the task of spectral band selection characterized by the relationship $x_i = c_i x_i'$, where $x_i$ represents the input of the $i$-th band, $c_i$ is a selection coefficient, and $x_i'$ denotes the original value

of the $i$-th spectral band. Let the importance measure $I_i$ be defined such that when $c_i$ is sparse and taking values in $\{0, 1\}$, $I_i$ corresponds to $c_i$ and thus can only take the values 0 or 1.

It follows that the subset $S_k$, comprising bands selected by the first $k$ $c_i$ coefficients, will be equivalent to the subset $T_k$, which is composed of the bands with the highest actual importance, $S_k = T_k$. Here, $k$ is the count of ones in the selection vector $c$.

## H.5 CONCLUSION

Based on Theorem 3, we understand that for any input of hyperspectral data $\mathbf{x}$ in a neural network, its output $\mathbf{f}(\mathbf{x})$ can be transformed into a weighted sum of each spectral band (Equation 33). This weighting, representing the network's implicit importance adjustment (or the implicit importance of spectral bands), is precisely what we aim to align with our importance module. According to Theorem 4, we know that sparsification is an effective way to align the band selection layer with the network's implicit importance. This theoretical understanding lends support to our method.

