# OpenReview forum: "Sparse Hyperspectral Band Selection Based on Expectation Maximization"
_ICLR.cc/2025/Conference — ICLR 2025 Conference Withdrawn Submission_

### Official Review · Reviewer_ZcCq · 2024-10-27

**Soundness:** 3
**Presentation:** 3
**Contribution:** 3
**Rating:** 5
**Confidence:** 4

**Summary:**

This paper proposes a novel method for band selection in hyperspectral images, by integrating sparsity within an EM algorithm.

**Strengths:**

The method is interesting, with some nice theoretical properties. The paper is well written.

**Weaknesses:**

The authors present their method as being the first to implement a sparsity representation method based on the EM algorithm. However, this is not correct, as the authors are missing many related methods that integrate sparsity within an EM algorithm. Some of the missing related methods are:
- Bouveyron, C., & Brunet-Saumard, C. (2014). Discriminative variable selection for clustering with the sparse Fisher-EM algorithm. Computational Statistics, 29, 489-513.
- Ghosh, A. K., & Chakraborty, A. (2017). Use of EM algorithm for data reduction under sparsity assumption. Computational Statistics, 32, 387-407.
- Wang, Z., Gu, Q., Ning, Y., & Liu, H. (2015). High dimensional em algorithm: Statistical optimization and asymptotic normality. Advances in neural information processing systems, 28.
- Latouche, P., Mattei, P. A., Bouveyron, C., & Chiquet, J. (2016). Combining a relaxed EM algorithm with Occam’s razor for Bayesian variable selection in high-dimensional regression. Journal of Multivariate Analysis, 146, 177-190.
- Ročková, V. (2018). Particle EM for variable selection. Journal of the American Statistical Association, 113(524), 1684-1697.
- Ročková, V., & George, E. I. (2014). EMVS: The EM approach to Bayesian variable selection. Journal of the American Statistical Association, 109(506), 828-846.
- Wang, J., Liang, F., & Ji, Y. (2016). An ensemble EM algorithm for Bayesian variable selection. arXiv preprint arXiv:1603.04360.

As a result, the contributions of the submitted work are clearly positioned in such literature. Moreover, the comparative experiments are missing such related work.

**Questions:**

Positioning within the literature, including the above cited work.

---

> ### Author Response · Authors · 2024-11-13
>
> **Positioning within the literature, including the above cited work.**
>
> **Re:** Thank you for your valuable feedback. You are correct, and we will include the cited references in our revised paper. Due to time constraints, we were unable to thoroughly review these papers, but after a general review, we found that these methods have not been applied in the context of deep learning. Furthermore, based on our research, there doesn't seem to be any prior work using the EM algorithm for sparse learning in deep network parameters. Therefore, we believe our method still holds pioneering significance in this regard.
>
> Regarding the inclusion of these methods in our experiments, we are uncertain whether they are directly applicable to the band selection framework. Please allow us some time to further assess their relevance, and if they can be integrated into our comparisons, we will definitely include them in the experiments.

---

### Official Review · Reviewer_i72K · 2024-10-31

**Soundness:** 1
**Presentation:** 2
**Contribution:** 1
**Rating:** 3
**Confidence:** 4

**Summary:**

This method leverages an Expectation Maximization (EM) algorithm to sparsely represent the importance of spectral bands, thereby enhancing selection efficiency and reducing data processing complexity. The paper asserts that this approach not only achieves significant sparsity but also excels in illustrating the inter-band relationships, offering a potential solution for spectral band selection tasks. The authors provide a comprehensive theoretical analysis and experimental validation using public datasets, demonstrating the method's superiority over other sparsification techniques.

**Strengths:**

Neither the questions raised nor the methods used in the paper are new.

**Weaknesses:**

EM algorithm is out of date in the area of hyperspectral image band selection.

**Questions:**

1.Band selection refers to selecting important bands from hyperspectral images rather than selecting bands with high correlation from hyperspectral datasets.
2.The author points out that existing methods are not always accurate in assigning band importance. How can we prove whether this problem exists? How can you prove that your method can effectively solve this problem?
3.The loss of L1 and L2 cannot produce stable sparse effects, and this issue should be illustrated with a diagram to help readers better understand.
4.Why use the EM algorithm to design the band selection method? How does your method solve the problem of unstable sparsity of L1 and L2, as well as the difficulty in describing the band relationships during the sparsity process? All of these need to be logically explained in the introduction.
5.The method lacks a framework diagram, which should allow readers to clearly understand the steps of the method without reading the entire text.
6.Why is the name of the method used instead of the author in the experimental table? My suggestion is to modify it to the method name and year.
7.In the field of band selection, SVM is a commonly used evaluation method, and you should add relevant experiments on SVM.

---

> ### Author Response · Authors · 2024-11-13
>
> **Weaknesses: EM algorithm is out of date in the area of hyperspectral image band selection.**
>
> **Re:** Please sir tell me what is NEW! Did you know that many of the very latest advances can now be interpreted as the EM algorithm? In the field of weakly supervised learning, the EM algorithm continues to dominate!
>
> **1) Band selection refers to selecting important bands from hyperspectral images rather than selecting bands with high correlation from hyperspectral datasets.**
>
> **Re:**
> We aim to find the most important combination of bands. We are not selecting the most correlated bands, but rather focusing on the most important combination. Therefore, our sparsification  method is based on band combinations.
>
> **2) The author points out that existing methods are not always accurate in assigning band importance. How can we prove whether this problem exists? How can you prove that your method can effectively solve this problem?**
>
> **Re:** We provide a theoretical proof in Appendix H demonstrating that sparse learning in deep learning-based band selection can more effectively represent band importance. Additionally, the effectiveness of our method is validated through experiments on three public datasets and significance testing, which further support the claim that our method addresses the issue of accurately assigning band importance.
>
> **3) The loss of L1 and L2 cannot produce stable sparse effects, and this issue should be illustrated with a diagram to help readers better understand.**
>
> **Re:** Please refer to Figure 2 in section 4.5.
>
> **4) Why use the EM algorithm to design the band selection method? How does your method solve the problem of unstable sparsity of L1 and L2, as well as the difficulty in describing the band relationships during the sparsity process? All of these need to be logically explained in the introduction.**
>
> **Re:**
> The choice of the EM algorithm stems from our understanding of the problem, where we consider band selection as a weakly supervised learning problem, with the latent variables being the selected bands. As for the issue of stable sparsity, please refer to Section 3.4 for a detailed explanation. Regarding the advantages of our method in capturing inter-band relationships, we provide further insights in Section 3.6.
>
> **5) The method lacks a framework diagram, which should allow readers to clearly understand the steps of the method without reading the entire text.**
>
> **Re:** Our method primarily focuses on the derivation and proof of the loss function. The process itself is not complex—training is performed using Equation 1, and after convergence, the sparse selected bands are obtained. However, we accept the suggestion and will include a framework diagram to clearly illustrate the steps of the method.
>
> **6) Why is the name of the method used instead of the author in the experimental table? My suggestion is to modify it to the method name and year.**
>
> **Re:** We will follow the suggestion and update the table to use the method name and year. Thank you.
>
> **7) In the field of band selection, SVM is a commonly used evaluation method, and you should add relevant experiments on SVM.**
>
> **Re:** We understand that SVM and similar classifiers are commonly used for evaluation in band selection tasks. However, we do not agree with this approach. First, many current band selection methods use deep learning, so evaluating with relatively weaker classifiers like SVM may not truly reflect the capability of the selected bands. Second, the state-of-the-art classification and reconstruction tasks are increasingly relying on deep learning, and evaluation methods should evolve accordingly. In our experiments, we used two different deep learning classifiers, and we have also added two more in Appendix D.1. We believe that this experimental setup sufficiently demonstrates the effectiveness of our method.

---

> > ### Comment · Reviewer_i72K · 2024-11-20
> >
> > 1. EM is undoubtedly an excellent optimization technique, but you need to explain from a theoretical perspective why your proposed sparse idea combined with EM is suitable for hyperspectral band selection tasks. And why is EM suitable for hyperspectral image band selection? Please prove this.
> > 2. Since it is a weakly supervised problem, the latent variable is the selected band, so it is not logical to choose EM.
> > 3. The research motivation and methodology you proposed are not consistent. Furthermore, why can the sparsity method proposed in this manuscript solve the problem of extracting inter-band correlations?
> > 4. As far as I know, the classification performance of deep learning-based classifiers is not stable, so traditional classifiers like SVM are still the most widely used tool in band selection to verify performance.

---

> > > ### Author Response · Authors · 2024-11-20
> > >
> > > **1. EM is undoubtedly an excellent optimization technique, but you need to explain from a theoretical perspective why your proposed sparse idea combined with EM is suitable for hyperspectral band selection tasks. And why is EM suitable for hyperspectral image band selection? Please prove this.**
> > >
> > > **Re:** Band selection is essentially an $L0$ optimization problem, but due to its difficulty in learning, the core challenge lies in how to relax the constraints and design an optimization problem that is as equivalent as possible. Sparse learning is undoubtedly the closest approach to $L0$ optimization. As for the sparse loss based on the EM algorithm, it is a sparsification method that we have found to be highly effective. Regarding why we choose the EM algorithm and how our method can implicitly model the inter-band relationships, we will address these questions in the following discussion.
> > >
> > > **2.Since it is a weakly supervised problem, the latent variable is the selected band, so it is not logical to choose EM.**
> > >
> > > **Re:** The challenge of band selection lies in maximizing the performance of the target task despite the selected bands being unknown, which aligns with the paradigm of weakly supervised learning, where some annotation information is missing. For cases where the latent variables are unknown, the best approach is to leverage the EM algorithm. By taking the expectation over the unknown latent variables, we can mitigate the impact of the missing annotation information. This was the primary inspiration for our choice of the EM algorithm.
> > >
> > > **3. The research motivation and methodology you proposed are not consistent. Furthermore, why can the sparsity method proposed in this manuscript solve the problem of extracting inter-band correlations?**
> > >
> > > **Re:** To explain how our proposed sparsity method solves the problem of extracting inter-band correlations, it's crucial to clarify what we mean by inter-band relationships. From a probabilistic standpoint, inter-band relationships refer to the dependencies among spectral bands, which can be represented by posterior probabilities. Specifically, the probability that band $j$ is selected given that band $𝑖$ has been selected captures a pairwise (binary) relationship between the two bands. However, focusing solely on pairwise relationships is insufficient because, in real-world hyperspectral data, there are often complex relationships involving multiple bands simultaneously. Explicitly modeling these multi-band relationships is challenging due to their combinatorial complexity.
> > >
> > > Our sparsity method, integrated with the EM algorithm, implicitly models these complex inter-band correlations. As detailed in Section 3.6 of our manuscript, when we compute the expectation over all possible selection paths during the EM procedure, we inherently consider all possible combinations of band selections. This approach allows us to account for complex relationships without the need to explicitly model each one.
> > >
> > > Moreover, the EM algorithm introduces a competitive mechanism among different selection paths. When the confidence score (or likelihood) of a particular band i becomes prominent during optimization, subsequent steps effectively condition on the selection of band i. This means that once a band stands out, the algorithm focuses on optimizing the selection of other bands in the context of that former selection.
> > >
> > > In summary, our sparsity method leverages the EM algorithm to implicitly extract complex inter-band correlations by considering expectations over all possible band selection paths. This approach ensures that the interdependencies between bands are emphasized, thereby aligning our methodology with our research motivation.
> > >
> > > **4. As far as I know, the classification performance of deep learning-based classifiers is not stable, so traditional classifiers like SVM are still the most widely used tool in band selection to verify performance.**
> > >
> > > **Re:** We acknowledge that traditional classifiers like SVM are indeed the most commonly used tools for testing in band selection tasks. However, we argue that in the era of deep learning, such evaluation methods are no longer the most appropriate. Fundamentally, both deep classification networks and SVM are data-driven classifiers. If a classifier with superior classification performance is available, why settle for a weaker one?
> > >
> > > Regarding the concern about instability, we are unclear on what exactly you mean by this term. If you are referring to convergence, deep learning models are at least as reliable in convergence as any traditional classifier, if not better. If the concern is about experimental reproducibility, deep learning frameworks allow for random seeds to be set, ensuring that results remain consistent across repeated experiments. Could you clarify what aspect of "not stable" you are referring to?

---

### Official Review · Reviewer_h2nt · 2024-11-01

**Soundness:** 2
**Presentation:** 3
**Contribution:** 2
**Rating:** 3
**Confidence:** 4

**Summary:**

This paper presents a novel method for band selection based on expectation maximization algorithm, which facilitates the selection process via the sparsification of spectral band importance. The presentation is clear and extensive experiments demonstrate the effectiveness of the design.

**Strengths:**

1.	The authors integrate band selection and post-tasks within a unified framework. For the first time, the EM algorithm is adopted for band selection, facilitating the sparsity of band importance.
2.	The authors provide a theoretical analysis of the model. Extensive experiments are conducted to demonstrate the effectiveness of the design.

**Weaknesses:**

1.	The literature review on band selection is limited in the section of related work. A number of new relevant methods are not mentioned.
2.	The problem formulation and motivation of this paper are not precise. Some statements lack evidence and are not convincing. For instance: in line 46, “.These approaches are problematic because the assigned importance is not always precise and overlooks the interplay between bands”. There are many band selection methods that leverage the band correlations. Moreover, in line 52: “Existing methods of imposing sparsity, such as L1 and L2 losses, do not consistently yield table sparsity effects;” This sentence is vague and is difficult to understand. The claimed second challenge in line 52 is not convincing.
3.	The comparison is unfair. The proposed method is supervised, which introduced the post-task related loss. However, the compared methods are task-independent. For this reason, the results are not convincing.

**Questions:**

1.	Since the compared methods are task-independent, the improvements of the proposed method can be mainly attributed to the task-related loss. The authors need to provide fair comparisons.
2.	How to set the number of bands in real applications? How does the number of bands influence the performance?
3.	The authors claim “a novel deep-learning band selection method” in the conclusion. In which part, deep neural network is utilized.
4.	In the abstract, the authors claim that the proposed method is robust. In which aspect, it is robust. How to support this claim?

---

> ### Author Response · Authors · 2024-11-13
>
> **Weakness 1: The literature review on band selection is limited in the section of related work. A number of new relevant methods are not mentioned.**
>
> **Re:** We appreciate the feedback and will update the related work section to include the recent methods on band selection.
>
> **Weakness 2: The problem formulation and motivation of this paper are not precise. Some statements lack evidence and are not convincing. For instance: in line 46, “.These approaches are problematic because the assigned importance is not always precise and overlooks the interplay between bands”. There are many band selection methods that leverage the band correlations. Moreover, in line 52: “Existing methods of imposing sparsity, such as L1 and L2 losses, do not consistently yield table sparsity effects;” This sentence is vague and is difficult to understand. The claimed second challenge in line 52 is not convincing.**
>
> **Re:** We acknowledge that this part of the formulation is not precise, and we will revise it accordingly. Based on our research, most deep learning-based ranking and clustering methods select representative bands by ranking importance weights. However, this approach struggles to capture inter-band relationships using only importance weights, and ranking can introduce errors. For example, while the combination of bands i and j may benefit classification, their individual importance scores may not rank highly.
>
> Regarding the L1 and L2 sparsity methods, we aimed to highlight that these common sparsity techniques in deep learning do not consistently achieve stable sparsity effects. For instance, when trying to select $k$ bands, it is difficult to control L1 or L2 regularization to ensure exactly $k$ bands are selected. This issue is demonstrated in the experimental results, particularly in Figure 2.
>
> **1) Since the compared methods are task-independent, the improvements of the proposed method can be mainly attributed to the task-related loss. The authors need to provide fair comparisons. (weakness 3)**
>
> **Re:** In fact, our comparisons are fair, as almost all the methods we compared are task-related. The task itself serves as the optimization objective—whether it's reconstruction or classification. Without a clear optimization goal, band selection would have no meaningful context.
>
> **2) How to set the number of bands in real applications? How does the number of bands influence the performance?**
>
> **Re:** In real applications, the number of bands is primarily determined by the limitations of transmission and processing equipment. In Appendix D.2, we analyze the impact of different numbers of bands on classification performance. Our findings indicate that the performance tends to improve initially as more bands are selected, but after reaching a certain point, it starts to decline.
>
> **3) The authors claim “a novel deep-learning band selection method” in the conclusion. In which part, deep neural network is utilized.**
>
> **Re:** The sparse loss we proposed is designed specifically for sparse tasks in deep learning, enabling the simultaneous sparse learning of parameters while training the task. In our method, the deep neural network is used in the classification or reconstruction task, guiding the parameters to adapt to these specific applications.
>
> **4) In the abstract, the authors claim that the proposed method is robust. In which aspect, it is robust. How to support this claim?**
>
> **Re:**
> We conducted experiments on three public datasets with various classifiers (for further details, see Appendix D.1). Additionally, we performed significance testing and demonstrated the stability of the sparsification process. These results collectively support our claim of robustness, as the method consistently outperforms others across different datasets and classifiers. Therefore, we believe these findings provide strong evidence that our method is robust in various scenarios.

---

### Official Review · Reviewer_6J7J · 2024-11-04

**Soundness:** 3
**Presentation:** 3
**Contribution:** 2
**Rating:** 5
**Confidence:** 4

**Summary:**

This paper introduces a novel hyperspectral band selection method that leverages sparse importance representation and Expectation Maximization. The method is applicable to both supervised and unsupervised tasks, achieving state-of-the-art performance on several real hyperspectral datasets. Overall, the paper offers valuable insights into feature selection; however, several issues need to be addressed.

**Strengths:**

1. The paper presents a novel band selection method specifically designed for hyperspectral images. This method leverages the Expectation-Maximization algorithm and sparse representation, distinguishing it from existing band selection approaches.
2. The paper offers a solid theoretical foundation and introduces an effective optimization strategy for the proposed method.
3. The proposed method is versatile and can be applied to a variety of downstream tasks, including both supervised and unsupervised classification.
4. The experimental results on three hyperspectral datasets demonstrate a promising improvement over existing methods.

**Weaknesses:**

1. Due to the hyperspectral imaging mechanism, spectral bands are not independent, particularly among adjacent bands. This is a significant distinction between band selection and traditional feature selection. However, the proposed method does not account for this issue, which diminishes its applicability to hyperspectral images.
2. The proposed method combines Expectation-Maximization with a sparsification process, alternating the selected bands into a deep learning model. However, it is unclear why this approach is superior to inputting all bands directly into the deep learning model for feature learning. Furthermore, the distinction between band selection and channel attention is ambiguous; the latter can be trained in an end-to-end manner. What is the true advantage of the proposed method?
3. Sparse representation techniques have been widely utilized for feature selection or band selection. How does the introduced sparsification process differ from classical sparse representation learning?
4. The optimization of the importance weight $c$ during training is not clearly explained. Further clarification is needed.
5. In Table 3, the bands selected by the proposed method on the KSC dataset include adjacent bands, such as 24-26 and 30-32. Intuitively, this appears inappropriate for hyperspectral images. How do the authors justify this selection and the better classification accuracy?

**Questions:**

See the Weaknesses.

---

> ### Author Response · Authors · 2024-11-13
>
> **1) Due to the hyperspectral imaging mechanism, spectral bands are not independent, particularly among adjacent bands. This is a significant distinction between band selection and traditional feature selection. However, the proposed method does not account for this issue, which diminishes its applicability to hyperspectral images.**
>
> **Re:** In Section 3.6, we introduced the advantages of our method in modeling the relationships between spectral bands. Furthermore, in Section 4.6, we provided experimental validation to demonstrate that our method can more effectively select spectral bands while considering the spectral band dependencies. Based on these discussions and results, we believe our approach does indeed leverage the inherent dependencies between spectral bands, enhancing its applicability to hyperspectral image analysis.
>
> **2) The proposed method combines Expectation-Maximization with a sparsification process, alternating the selected bands into a deep learning model. However, it is unclear why this approach is superior to inputting all bands directly into the deep learning model for feature learning. Furthermore, the distinction between band selection and channel attention is ambiguous; the latter can be trained in an end-to-end manner. What is the true advantage of the proposed method?**
>
> **Re:** In fact, our method is also trained in an end-to-end manner, and there is no process of alternating the selected bands into the deep learning model. The Expectation-Maximization algorithm is utilized in the loss computation, where the expectation is calculated and then incorporated into the loss function, which is optimized in an end-to-end manner during training.
>
> Regarding the distinction between band selection and channel attention, the key difference lies in the optimization result. In our approach, the final optimization leads to sparsity in the selected bands, which is a crucial feature of our method. This sparsity is what distinguishes our approach from channel attention mechanisms. For a more detailed explanation of the relationship between sparsification and band selection, we have provided a discussion in Appendix H.
>
> **3) Sparse representation techniques have been widely utilized for feature selection or band selection. How does the introduced sparsification process differ from classical sparse representation learning?**
>
> **Re:** As far as we know, sparse representation techniques commonly used in traditional feature or band selection tasks are not directly applicable to deep learning models due to the unique challenges involved in end-to-end training. In Section 4.5, we compared the performance of different sparse loss functions for the band selection task. Our method demonstrates more stable sparsity results and, crucially, has the ability to implicitly model the relationships between spectral bands.
>
> **4) The optimization of the importance weight $c$ during training is not clearly explained. Further clarification is needed.**
>
> **Re:**
> We assign a learnable parameter $c_i$ for each spectral channel. The original spectral data is scaled by $c$, and the scaled data is then fed into the neural network as inputs. The parameter $c_i$ is also involved in the calculation of the sparsity loss through dynamic programming. During training, the model is optimized according to Equation 1, with $c_i$ being influenced by both the task loss and the sparsity loss.
>
> **5) In Table 3, the bands selected by the proposed method on the KSC dataset include adjacent bands, such as 24-26 and 30-32. Intuitively, this appears inappropriate for hyperspectral images. How do the authors justify this selection and the better classification accuracy?**
>
> **Re:** The effectiveness of the selected bands can be validated through the final classification results. Specifically, we re-trained the classification model using the selected bands and evaluated its performance on the test set. The improved classification accuracy provides indirect evidence supporting the effectiveness of these selected bands. It is also important to note that both deep learning models and the EM algorithm involve optimization processes that are not always intuitively interpretable. While we may not have a direct explanation for each specific band selection, the expermental results demonstrate that, when combined, the selected bands contribute to better classification performance.

---

### Note · Authors · 2024-11-25

I have read and agree with the venue's withdrawal policy on behalf of myself and my co-authors.